# Measuring SDG 15 at the County Scale: Localization and Practice of SDGs Indicators Based on Geospatial Information

**Shaoyang Liu [1], Jianjun Bai [1],\* and Jun Chen [2]**

[1]   School of Geography and Tourism, Shaanxi Normal University, Xi'an 710119, China; liushaoyang@snnu.edu.cn
[2]   National Geomatics Center of China, Beijing 100830, China; chenjun@nsdi.gov.cn
\*   Correspondence: bjj@snnu.edu.cn

**Abstract:** To achieve the goal of worldwide sustainable protection and utilization of terrestrial ecosystems, it is necessary to quantitatively assess the implementation of Sustainable Development Goal 15 (SDG 15) at all administrative levels, especially at the grass-roots level, using the indicator framework of the UN SDGs. However, in the SDG 15 indicator system, the relationship between goal and indicators is ambiguous, and the results of the indicators cannot be visualized to show the differences within regions. Moreover, its design scale is country-oriented, which suggests that the indicator system cannot be applied directly to the county levels. In light of these issues, this paper used four modalities of localization to form an indicator system of localization, and applied it in the quantitative evaluation of meeting the objectives of SDG 15 in Deqing County, China. The localized indicator system for county level based on geospatial information included six indicators, which were clustered into three categories: sustainable forest management, halt and reverse land degradation, and conservation of biodiversity. By comparing and evaluating the quantitative results of SDG 15 in Deqing, 70% of the comparable indicators in the localization indicator system were at the forefront of those in China or the world. The results showed that grouped analysis of the targets and indicators could clarify the relationship between the implications of the goal and indicators, and the indicator system based on the geographic information was conducive to displaying the spatial distribution of the results of the indicators and clarifying the internal differences.

**Keywords:** SDG indicators; localization; geospatial information; county scale; Deqing County

## 1. Introduction

To promote the coordinated development of the trio of economic growth, social inclusion and environmental sustainability worldwide, the 17 Sustainable Development Goals (SDGs) from the 2030 Agenda for Sustainable Development were adopted by world leaders in September 2015 at an historic UN Summit [1]. The United Nations further proposed and promoted a systematic follow-up and review of the implementation of this global agenda, including quantitative assessment and reporting of the progress towards the SDGs through comprehensive utilization of statistical and geographic information methods [2]. The United Nations has established an Inter-Agency and Expert Group on the SDG indicators (IAEG-SDGs) and conducted research on the indicator design, metadata compilation, indicator classification, etc. In 2017, the SDGs Global Indicator Framework (SGIF), which included 232 indicators, was proposed [3], providing a globally unified indicator system for quantitative assessment, periodic monitoring, and reporting of the national or regional SDGs.

After establishing the SGIF, many countries and scholars have discussed the design, quantification and application of the indicator system. Some studies have clearly pointed out or indirectly reflected the

shortcomings and limitations of the SDGs indicator system. First, in terms of the connotation analysis and framework design of the SDGs, some studies have discussed the analysis and reconstruction of the indicators [4,5], the relationship between the SDGs [6], and the construction of the core variables and framework [7]. They reflected the lack of relevance between the goals and indicators, and the inadequate applicability of the SGIF. Second, some countries and organizations have used statistical information to conduct comprehensive evaluation and monitoring of the SDGs [8], but they have not fully utilized geospatial information, so the evaluation results are unable to reflect the geospatial patterns, regional differences and spatiotemporal effects [9]. In practice, because the indicator framework was designed to be used at a national level, it is difficult to apply it at within-national levels [10]. However, the results of evaluations of the small regions, especially county-scale administrative regions, would have more practical significance for local government action.

These shortcomings and limitations of the SDGs indicator system also exist in Sustainable Development Goal 15 (SDG 15), which reads "Protect, restore and promote sustainable use of terrestrial ecosystems, sustainably manage forests, combat desertification, and halt and reverse land degradation and halt biodiversity loss".

### 1.1. The Shortcomings of the Indicator Framework Design

The indicator framework of SDG 15 contains 12 targets and 14 indicators (Appendix A). The development of the indicator system was not a one-step process. Until now, its conceptual design and the computing methods of indicators are still worth discussing. There are some problems in the design of the framework, such as the duplication of indicators, broad definitions, and unclear grouping. Some experts believe that the number of SDG indicators is too high, however the overall description of the goals and targets is insufficient and lacks essential indicators (essential sustainable development variables that capture major dimensions of the change in various systems and define a minimum core set of social, environmental, and economic measurements for SDGs monitoring) [11]. In addition, the indicator method designed by IAEG-SDG, does not divide the goal into groups [12]. The indicators for SDG 15 are complex, and the targets are ineffectively correlated. Indicators 15.1.2 (Proportion of important sites for terrestrial and freshwater biodiversity that are covered by protected areas, by ecosystem type) and 15.4.1 (Coverage by protected areas of important sites for mountain biodiversity) both express "the proportion of biodiversity important sites covered by protected areas", but they belong to two different targets. Indicators 15.2.1 (Progress towards sustainable forest management) and 15.3.1 (Proportion of land that is degraded over total land area) are more broadly defined and need further division. Multiple indicators even overlap under different targets. Indicators 15.7.1 (Proportion of traded wildlife that was poached or illicitly trafficked) and 15.c.1 (Proportion of traded wildlife that was poached or illicitly trafficked), 15.a.1 (Official development assistance and public expenditure on conservation and sustainable use of biodiversity and ecosystems) and 15.b.1 (Official development assistance and public expenditure on conservation and sustainable use of biodiversity and ecosystems) completely overlap in their definitions and calculation methods, but targets 15.7 and 15.c, 15.a, and 15.b have different emphases. Second, the existing metadata descriptions of the indicators fail to cover all indicators, and some indicators lack clear definitions. Indicator 15.9.1 (Progress towards national targets established in accordance with Aichi Biodiversity Target 2 of the Strategic Plan for Biodiversity 2011–2020) shows the "progress of national goals", which is difficult to describe with a specific index; no data for Indicator 15.9.1 is currently available, and its methodology is still under development.

### 1.2. The Limitations of Indicators in Visualizing Spatial Variability

In the preamble to Agenda 2030, it was stated that "we pledge that no one will be left behind" in the process of achieving sustainable development [1]. This promise requires the review of every detail of the regional progress towards meeting the SDGs, but the existing SDGs monitoring reports use numerical values to show the state of indicators and goals, as does the United Nations Sustainable Development

Goals Report. However, regional averaging often conceals spatial variability, and subdividing the data according to multiple dimensions, including age, gender, and geographical location, is critical for ensuring that no one is left behind [13]. Since 2011, the geospatial community, working closely with the statistical community, has investigated how geospatial information can be used to improve the production of many SDG indicators by establishing the United Nations Committee of Experts on Global Geospatial Information Management [14]. However, of the 232 indicators in the SDGs indicator system, only approximately 24 indicators can be visually displayed using geospatial information. In SDG 15, the number of indicators (6/14, 43%) that can be directly or indirectly calculated and expressed by geospatial information is the highest of the 17 2030 UN SDGs. The majority of the indicator calculations only use social and economic statistics, resulting in a numerical value. Indicators 15.1.1 (Forest area as a proportion of total land area), 15.1.2 (Proportion of important sites for terrestrial and freshwater biodiversity that are covered by protected areas, by ecosystem type), 15.2.1 (Progress towards sustainable forest management), 15.3.1 (Proportion of land that is degraded over total land area), 15.4.1 (Coverage by protected areas of important sites for mountain biodiversity) and 15.4.2 (Mountain Green Cover Index) are related to geospatial information, but because the spatial visualization of Indicator 15.2.1 (Progress in sustainable forest management) and 15.3.1 (The proportion of degraded land to total land area) are unclear, further deconstruction and analysis of its definition and methods is needed. Displaying the spatial details and differences in the results of the SDG 15 indicators could allow the results of the quantitative evaluation and the proposed policy-oriented control or planning measures to be implemented in specific spatial locations [10].

*1.3. The Shortcomings of the Scale Applicability of the Indicators Framework*

The United Nations pointed out that "Indeed, the rate of global progress is not keeping pace with the ambitions of the Agenda, necessitating immediate and accelerated action by countries and stakeholders at all levels" [14]. Although the SDGs are an important achievement in many respects, the basic issues remain how to monitor and evaluate their implementation at the global and most direct local levels [12]. Furthermore, it is commonly believed that the SDGs cannot be achieved by the actions of governmental or intergovernmental actors but necessitate the involvement of decision-makers at all levels, particularly those operating at the grassroots-level [15]. However, the existing SDGs indicator framework was proposed at a global scale and cannot be directly applied to countries at different stages of development and regions at different scales. When SDG 15 is applied at the county scale, more than 60% of the indicators cannot be directly applied. In the indicator system, Indicators 15.6.1 (Number of countries that have adopted legislative, administrative and policy frameworks to ensure fair and equitable sharing of benefits), 15.8.1 (Proportion of countries adopting relevant national legislation and adequately resourcing the prevention or control of invasive alien species) and 15.9.1 (Progress towards national targets established in accordance with Aichi Biodiversity Target 2 of the Strategic Plan for Biodiversity 2011–2020) calculate the number and progress of countries; Indicators 15.7.1 (Proportion of traded wildlife that was poached or illicitly trafficked), 15.a.1 (Official development assistance and public expenditure on conservation and sustainable use of biodiversity and ecosystems), 15.b.1 (Official development assistance and public expenditure on conservation and sustainable use of biodiversity and ecosystems) and 15.c.1 (Proportion of traded wildlife that was poached or illicitly trafficked) measure the national-level results in the definition and calculation methods; the applicable scales of the reference database sources provided in the metadata documents 15.1.2 and 15.4.1 require further discussion and research. Indicators should be localized to form a localized indicator system that aligns with local conditions [10] so that the decision makers in local government can obtain detailed and targeted reference opinions when the corresponding strategies or improvement measures are implemented at specific county-scale regions.

Taking SDG 15 as a research objective, this paper aimed to localize an indicator system, and use it to quantify county-level progress towards the objectives set in SDG 15. Furthermore, the limitations of the SDG 15 indicator system are analyzed and discussed. We fully utilized the geospatial information

to improve the indicators and to construct a county-scale localized SDG indicator system according to the small-scale spatial characteristics. This study provides a direct and accessible calculation method for the future county-scale evaluation of SDG 15 and a research idea and reference basis for further improvement of the SDG indicators.

## 2. Study Area and Data

### 2.1. Study Area

The study area chosen in this paper was Deqing County, Zhejiang Province, China. In the past 40 years, the economy has developed rapidly, and the level of modernization has been increasing in China. Simultaneously, the county has also noted changes in the ecological environment and taken action to create an environmentally friendly society. Deqing is the epitome of these changes in China, including the economic and environmental changes. In recent years, with the rapid development of the economy, emerging industries that employ geographic information have developed in Deqing, and the environment, economy and society and their relationships have changed dramatically; therefore, a suitable evaluation system is urgently needed to monitor its sustainable development process.

Deqing County is located eastern China (Figure 1). The eastern parts of Deqing County have low terrain, while western parts have high terrain in. The average elevation is 75 m and the total area is 937.92 km$^2$. The area has a subtropical humid monsoon climate with average annual temperatures of 13 °C to 16 °C and an average annual precipitation of 1379 mm. The vegetation type is evergreen broad-leaved forest in the middle subtropical zone, and approximately fifty percent of the woodland is composed of bamboo plants. Deqing has a high-quality ecological environment, which is suitable for the survival of wild animals and plants. After the extinction of the crested ibis in Zhejiang in the last century, the ecological environment has improved, and crested ibis can again reproduce and flourish in Deqing County, Zhejiang Province.

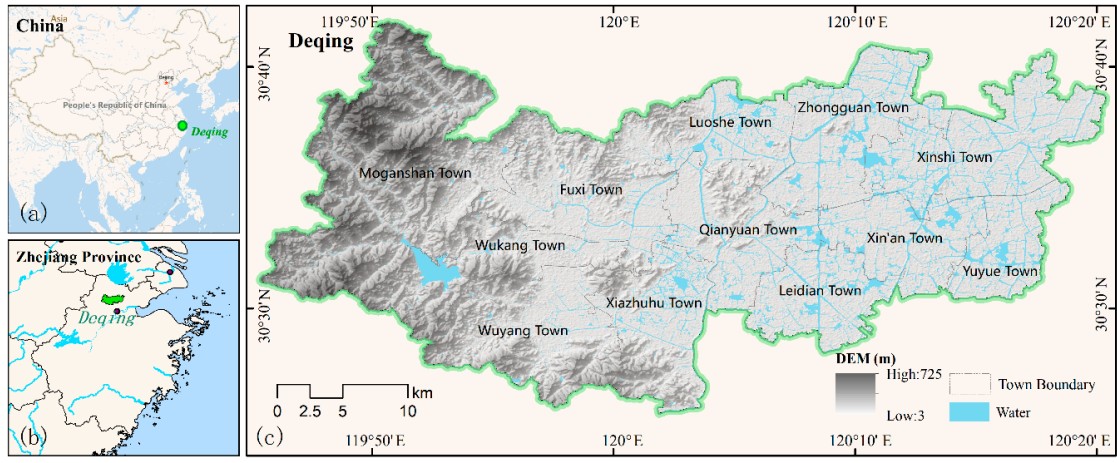

**Figure 1.** Geographical location of Deqing: (**a**) Location of Deqing County in China; (**b**) Location of Deqing County in Zhejiang Province; (**c**) Geography of Deqing County.

### 2.2. The Data Used in this Study

(1)    Landsat images with a spatial resolution of 30 m were used in this study. These medium-resolution remote-sensing data are able to distinguish basic land cover types.

Multi-spectral remote sensing images from the Landsat 7 ETM+ and Landsat 8 OLI sensors from 2012 to 2017 were used; the image phase was the April to November vegetation growing season, for the extraction of forests and the differentiation of the natural vegetation from artificial surfaces. For the

six-year monitoring period, 13 images with cloud cover less than 30% were selected, with at least one image per year.

(2)    Vector data of the surface coverage in 2015 and aerial photographs with spatial resolutions of 0.5 m from 2012 to 2017 were provided by the Geomatics Center of Deqing County, and were used to select training samples for image interpretation and classification (forest extraction) and to verify the classification conducted using Landsat data.

(3)    Data of the daily temperature and precipitation at the meteorological stations were provided by Meteorological Bureau of Deqing County from 2012 to 2017, and there were 12–22 meteorological stations (new stations will be added every year) in Deqing County; data about the solar radiation stations were downloaded from National Meteorological Information Center of China.

(4)    DEM data of ASTER GDEM were used for land cover classification as the terrain feature parameters.

## 3. Analysis of the Connotation of SDG 15 and Methodology

### 3.1. Analysis and Clustering of the Connotation of SDG 15

Before the localization of the indicators, we needed to examine the connotation of the goal in detail and clarify what the goal intends to achieve.

The theme of SDG 15 is the sustainable utilization of terrestrial ecosystems; it aims to "protect, restore and promote sustainable use of terrestrial ecosystems". Its targets and indicators follow the theme of goal, serving the three aspects: sustainable forest management, halt and reverse land degradation and conservation of biodiversity. Targets 15.1, 15.2, 15.4 and 15.b focus on the state of terrestrial ecosystems, with emphasis on monitoring and management of forests and vegetation; Target 15.3 directly indicates the status of land degradation; the other seven targets, including 15.1 and 15.4, emphasize biodiversity conservation in terms of important sites for biodiversity conservation, species abundance, management and financial support. Strengthening forest resource management, combating land degradation and desertification and protecting biodiversity are three important avenues for protecting, restoring and promoting the sustainable use of terrestrial ecosystems.

Through the analysis of the connotation of SDG 15, we divided its targets into three groups (Table 1) to reflect the three connotations of this goal, namely, sustainable forest management, halt and reverse land degradation, and conservation of biodiversity.

(1)    Sustainable forest management (SFM)

SFM is a dynamic and evolving concept that aims to maintain and enhance the economic, social and environmental values of all types of forests, for the benefit of present and future generations (Resolution A/RES/62/98, UN General Assembly).

(2)    Halt and reverse land degradation

In combating and reversing land degradation, the goal focus on the proportion of degraded land to monitor the situation of combating desertification and restoring degraded land. Land degradation refers to the deterioration of land cover, the aggravation of soil erosion, the thinning of the soil layer, the decrease in land productivity, the decrease in the environmental capacity for the population and the plunge of the ecosystem into a vicious circle caused by the unreasonable use of land by human beings in the fragile ecological environment. The prevention and reversal of land degradation and the maintenance and stabilization of land productivity are of great significance for ensuring ecosystem stability and food security.

**Table 1.** Connotation clustering of Sustainable Development Goal (SDG) 15 indicator system.

| Content | Targets | Indicators |
|---|---|---|
| Sustainable forest management | 15.1 Protection, restoration and sustainable use of terrestrial and freshwater ecosystems and their services | 15.1.1 |
| | 15.2 Promoting sustainable forest management and increasing afforestation | 15.2.1 |
| | 15.4 Protecting Mountain Ecosystems, including their biodiversity | 15.4.2 |
| | 15.b Mobilize significant resources from all sources and at all levels to finance sustainable forest management | 15.b.1 |
| Halt and reverse land degradation | 15.3 Combat desertification, restore degraded land and soil | 15.3.1 |
| Conservation of biodiversity | 15.1 Protection, restoration and sustainable use of terrestrial and freshwater ecosystems and their services | 15.1.2 |
| | 15.4 Protecting Mountain Ecosystem and Biodiversity | 15.4.1 |
| | 15.5 Protect and prevent the extinction of threatened species | 15.5.1 |
| | 15.6 Promote fair and equitable sharing of the benefits arising from the utilization of genetic resources | 15.6.1 |
| | 15.7 End poaching and trafficking of protected species of flora and fauna | 15.7.1 |
| | 15.8 Reduce the impact of invasive alien species | 15.8.1 |
| | 15.9 Integrate ecosystem and biodiversity values into national and local planning | 15.9.1 |
| | 15.a Mobilize and significantly increase financial resources from all sources to conserve and sustainably use biodiversity and ecosystems | 15.a.1 |
| | 15.c Enhance global support for efforts to combat poaching and trafficking of protected species, | 15.c.1 |

(3)　Conservation of biodiversity

The conservation of biodiversity focuses on the conservation of important sites and species, as well as progress of the measures taken to curb the loss of biodiversity. Biodiversity is the basis for the survival and development of humans and provides indispensable biological resources for human survival.

### 3.2. Localization Reform of SDG 15

The global indicator system has a wide-ranging scope and is suitable for national SDG monitoring. A localization process must be undertaken when applying the framework at a sub-national or regional level (such as in Deqing County). The indicators are selected to align with sub-national (or regional) geographical circumstances. While some indicators can be selected or adopted directly from the global framework, others might need to be revised or extended.

The reform of indicator systems should follow some basic principles based on the purpose and applicability of the reformed indicators. One of the core ideas of localization reform is to be able to understand and accurately reflect the connotation and themes of SDG goals. It is necessary to analyze the connotation and clustering of the indicators, to modify and supplement the uncertainties in the setting and calculation of indicators and to create a trade-off between the unsuitable places. In addition, data availability and reliability are other important factors to consider. The geographic information data and methods should be fully used to visually display the spatial details of indicator results. The workflow of localization reform and application of SDG 15 is shown in Figure 2.

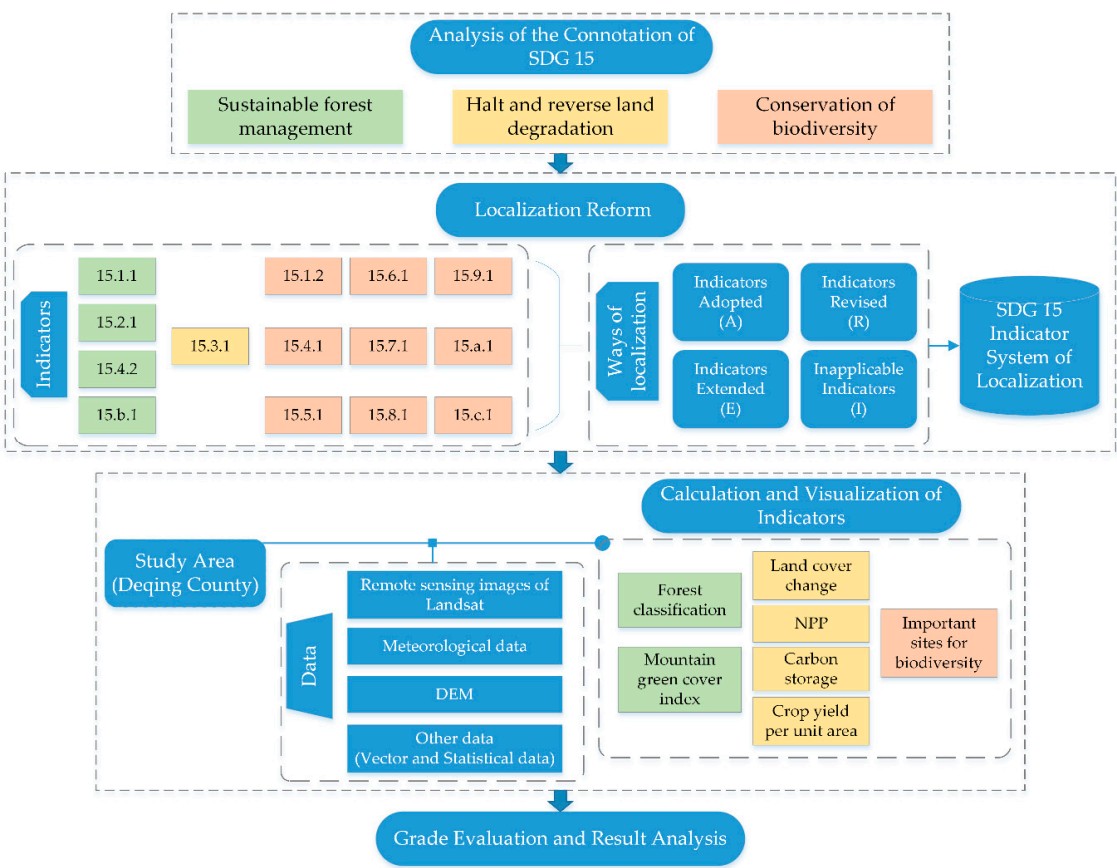

**Figure 2.** Workflow diagram.

We classified the methods of localization into four categories.

(1)　Indicators Adopted (A)

Without changing the original indicator name, definition and calculation method, the indicator can be directly used.

- Indicator 15.1.1: Forest area as a proportion of total land area, which is calculated as follows:

$$F = \frac{A(forest)}{A(total)} \tag{1}$$

where: *F* is the forest area as a proportion of the total land area, *A(forest)* is the forest area, and *A(total)* is the total administrative land area.

In this study, 13 multispectral remote sensing images were used to classify land cover types and extract forest area using a machine learning method (random forest). The random forest package in R Studio was used to construct the random forest model. Four kinds (spectral, exponential, texture, and topographic features) of 70 characteristic variables were extracted. In the process of texture feature extraction, there is data redundancy when extracting features from all bands of images. Principal component analysis (PCA) is a method that can be used to remove the redundant information between the bands and compress the multi-band image information to a few more effective conversion bands than the original band. We used PCA to calculate the first and second principal components, which contained at least 90% of the information for all bands and extracted the texture features from the first and second principal components. Then, we used the random forest importance score; according to the importance score, we selected the top 45 (after the experiment, when the number of features was 45, the classification accuracy was the highest) feature variables as the input for the classification model.

- Indicator 15.4.2: Mountain Green Cover Index

The Green Cover Index is intended to measure the changes of the green vegetation in mountain areas—i.e., forest, shrubs, trees, pasture land, crop land—to monitor progress on the mountain target.

The normalized difference vegetation index (NDVI) based on the pixel dichotomy model was used to calculate the Green Cover Index in mountainous areas as follows:

$$MGCI = \frac{NDVI_x - NDVI_{soil}}{NDVI_{veg} - NDVI_{soil}} \tag{2}$$

where: *MGCI* is the Mountain Green Cover Index; $NDVI_x$ is the NDVI of the pixel *x*; $NDVI_{soil}$ is the NDVI of the pure bare soil pixel in the study area; and $NDVI_{veg}$ is the NDVI of the pure vegetation pixel in the study area.

(2) Indicators Extended (E)

The name, definition and calculation method of original indicator can be quoted, and then can be divided into other sub-indexes to supplement.

- Indicator 15.3.1: Proportion of land that is degraded over total land area

Land degradation is defined as the reduction or loss of the biological or economic productivity and complexity of the rain fed cropland, irrigated cropland, or range, pasture, forest and woodlands resulting from a combination of pressures, including land use and management practices.

It contains three sub-indicators as follows:

(a) Evaluation of land cover and land cover change;
(b) Land productivity status and trend analysis based on net primary production;
(c) Measure carbon reserves and changes.

The above three sub-indicators show the state of the natural environment and the soil layer as a whole. The most direct manifestation of land degradation is the change in vegetation, particularly, crop yield, which is reflected in human life and the economy. Therefore, the indicator of "d). crop yield per unit area" has been added to reflect the state of cultivated land productivity.

(a). According to the metadata description "Land cover refers to the observed physical cover of the Earth's surface which describes the distribution of vegetation types, water bodies and human-made infrastructure. It also reflects the use of land resources (i.e., soil, water and biodiversity) for agriculture, forestry, human settlements and other purposes". Land cover types are divided into two systems: natural surface, including woodland, grassland, cultivated land and water surface, and artificial surface, including human settlements, and unused land. This sub-indicator shows the distribution of natural surfaces and artificial surfaces and their mutual changes. The direction of land degradation is defined as the transformation of a natural surface into an artificial surface as follows:

$$P_{i,n} = \frac{A(degraded)_{i,n}}{A(total)_{i,n}} * 100\%$$
(3)

where: $P_{i,n}$ is the proportion of degraded land in the land cover class $i$ in the year of monitoring $n$; $A(degraded)_{i,n}$ is the area of land degradation in the land cover class $i$ in the year of monitoring $n$; and $A(total)_{i,n}$ is the total area of the land cover class $i$.

We use change detection matrix to analyze the land cover change.

(b). Based on Carnegie Ames-Stanford Approach (CASA) model, net primary production (NPP) was calculated using land cover type data, meteorological data, and solar radiation data.

The calculation principle of CASA model [16] is as follows:

$$NPP(x,t) = APAR(x,t) \times \varepsilon(x,t)$$
(4)

$$\varepsilon(x,t) = \varepsilon_{max} \times T_\varepsilon(x,t) \times W_\varepsilon(x,t)$$
(5)

$$APAR(x,t) = SOL(x,t) \times 0.5 \times FPAR(x,t)$$
(6)

$$FPAR(x,t) = \frac{(NDVI(x,t) - NDVI_{i,min}) \times (FPAR_{max} - FPAR_{min})}{NDVI_{i,max} - NDVI_{i,min}} \times FPAR_{min}$$
(7)

where $NPP(x,t)$ is the net primary productivity of the vegetation in pixel x in time t; $APAR(x,t)$ is the absorbed photosynthetic active radiation; $\varepsilon(x,t)$ is the actual utility rate of luminous energy; $T_\varepsilon(x,t)$ is the coefficient of temperature stress; $W_\varepsilon(x,t)$ is the coefficient of water stress; $\varepsilon\_max$ is the theoretical maximum utility rate of the luminous energy reaching the vegetation; $SOL(x,t)$ is the total solar radiation (MJ/m$^2$); $FPAR(x,t)$ is the photosynthetic effective radiation absorptivity; 0.5 is the ratio of solar effective radiation absorbed by vegetation (wavelength 0.38–0.71 μm) to total solar radiation received by vegetation canopy; and $NDVI$ is the normalized difference vegetation index.

(c). The Integrated Valuation Ecosystem Services and Tradeoffs (InVEST) model was used to calculate carbon storage. The model uses distributed algorithm based on 3S technology to solve the limitations of traditional assessment methods. It is a new technology for spatial expression, dynamic analysis and quantitative assessment of ecosystem services.

The carbon storage assessment module of InVEST includes four carbon pools, i.e., aboveground carbon, underground carbon, soil carbon and dead biological carbon. Carbon storage on a piece of land largely depends on the size of the four carbon pools [17].

$$C_v = C_{above} + C_{below} + C_{soil} + C_{dead}$$
(8)

where $C_v$ is the total carbon storage, $C_{above}$ is the aboveground carbon, $C_{below}$ is the underground carbon, $C_{soil}$ is the soil carbon, $C_{dead}$ is the carbon of dead organisms (litter, etc.).

The parameters of the carbon pool used in the model were taken from the relevant literature [18–20]. According to the land cover type in this paper, the corresponding parameters were obtained directly from the literature, and the land cover without corresponding type was synthesized from similar land use types.

(3)   Indicators Revised (R)

In view of the interpretation of the indicators in the UN metadata documents and considering the application scale and research purposes, if the content of indicators cannot be calculated, they would be abandoned or modified to usable indicators with similar meanings. If the data source was unavailable or the scale was not applicable, the available data should be used to construct a new usable dataset.

- Indicator 15.1.2: Proportion of important sites for terrestrial and freshwater biodiversity that are covered by protected areas, by ecosystem type

In this indicator, the UN SDGs metadata recommend the use of the World Database on Key Biodiversity Areas (WDKBA) under A Global Standard for the Identification of Key Biodiversity Areas [21], prepared by the International Union for Conservation of Nature (IUCN), for the data of Biodiversity Important Sites. However, in the IUCN standard, the survey and evaluation scale of key biodiversity areas (KBAs) is at a national or global scale, and the data in the WDKBA established according to the standard are not suitable for small-scale areas below national scale. This is because, in some subsets of the national-scale data (spatial polygon data), it is very likely that they contain small fragmented disturbance patches (such as construction land, hardened roads, bare land and other landscapes that may have a negative impact on biodiversity); locations that are not in the database may not necessarily have low biodiversity but may be of high relative importance for biodiversity in a certain area of the region. Moreover, the KBAs in the WDKBA were classified according to the survey data of the species richness at each threat level in the region. In some countries lacking species and biological resources inventory, such databases cannot be established and it is difficult to make consistent and accurate measurements. In fact, traditional environmental data have long suffered from data breaks, due to changes in reporting methods and from data gaps [22].

Therefore, another important step for us is to study the remote sensing identification method for identifying the KBAs, which provides a set of convenient and available remote sensing data sets, and effectively solves the scale limitation of the WDKBA metadata.

Considering the connotation of the different levels of biodiversity, we selected five sub-indicators that were most closely related to the biodiversity at each level and constructed a biodiversity index (BI) based on remote sensing data to identify important areas for biodiversity. There are four levels of biodiversity: genetic diversity, species diversity, ecosystem diversity, and landscape diversity. At the level of species diversity, we considered habitat quality (habitat quality index), productivity status (net primary productivity), and vegetation index (enhanced vegetation index, EVI); at the level of ecosystem diversity, the proportion of habitat (biotope) area (percentage of habitat area) was taken into account; at the level of landscape diversity, Shannon's diversity index (SHDI) was used to evaluate the heterogeneity and diversity of landscape. Since the evaluation of genetic diversity involves statistical data at the gene level and for specific species, it cannot be directly monitored through remote sensing; the individuals of different species have diverse genes, so genetic diversity can be reflected in species diversity [23], and there are no indicators of the levels of genetic diversity level. The SDG 15.1.2 metadata description refers to terrestrial and freshwater ecosystems, but the content and methods of biodiversity assessment of terrestrial and freshwater ecosystems are different. At this stage, this study only focused on terrestrial ecosystems.

The BI is calculated as follows:

$$BI = HQI \times \beta_1 + NPP \times \beta_2 + EVI \times \beta_3 + S_p \times \beta_4 + SHDI \times \beta_5 \tag{9}$$

where: $HQI$ is the habitat quality index; $NPP$ is the net primary productivity; $EVI$ is the enhanced vegetation index; $S_p$ is the percentage of habitat area; $SHDI$ is the Shannon's diversity index; and $\beta_i$ is the weight of each sub-indicator (Table 2).

- Indicator 15.2.1: Progress towards sustainable forest management

The indicator is composed of five sub-indicators as follows:

1.   Forest area annual net change rate;
2.   Above-ground biomass stock in forest;
3.   Proportion of forest area located within legally established protect areas;
4.   Proportion of forest area under a long-term forest management plan; and
5.   Forest area under an independently verified forest management certification scheme.

**Table 2.** The indicator system and weight for Biodiversity Index.

| Biodiversity Level | Sub-Indicators | Weight |
|---|---|---|
| Species diversity | Habitat quality index (HQI) | 0.35 |
| | Net primary productivity (NPP) | 0.25 |
| | Enhanced vegetation index (EVI) | 0.15 |
| Ecosystem diversity | Percentage of habitat area ($S_p$) | 0.10 |
| Landscape diversity | Shannon's diversity index (SHDI) | 0.15 |

Note: the weights of five sub-indicators are determined by the analytic hierarchy process (AHP).

Among the sub-indicators, there are no uniform fixed criteria or scope, and many areas of sustainable forest management are not certified either because their owners choose not to seek certification or because there is no credible or affordable certification scheme. Therefore, these two sub-indicators are difficult to quantify, so they were not adopted. In addition, due to the lack of necessary data, it was temporarily impossible to calculate "2. Above-ground biomass stock in forest".

• Indicator 15.4.1: Coverage by protected areas of important sites for mountain biodiversity

This indicator emphasizes the distribution and change of important sites for biodiversity in mountainous areas. The identification of important sites for biodiversity in mountainous areas was the same as the Indicator 15.1.2.

(4)   Inapplicable Indicators (I)

The following indicators, either due to the use of statistical data and survey data alone, through the measurement the numbers or plans of countries, or due to the lack of metadata, and make the evaluation of these indicators on small spatial scales meaningless or inappropriate.

• Indicator 15.5.1: Red List Index

The Red List Index calculates the endangerment of all species in the region. In fact, most species exist in a wide range, usually within the boundaries of the whole ecological area (habitat). However, the boundaries of county administrative areas are not defined according to the habitat of species, and small-scale administrative areas often divide the living areas of species. Therefore, at the county level, the species and numbers of animals and plants are variable. An index cannot significantly reflect the endangered status of a species in a county. When bounded by the country or region, an indicator can play a better role.

• Indicator 15.6.1: Number of countries that have adopted legislative, administrative and policy frameworks to ensure fair and equitable sharing of benefits

This indicator calculates the number of countries, which is meaningless at the county level.

• Indicator 15.7.1: Proportion of traded wildlife that was poached or illicitly trafficked

This indicator calculates the share of international trade among countries, and it is difficult to quantify the types of data in county-based administrative regions.

• Indicator 15.8.1: Proportion of countries adopting relevant national legislation and adequately resourcing the prevention or control of invasive alien species

This indicator calculates the number of countries, which is meaningless at the county scale.

- Indicator 15.9.1: Progress towards national targets established in accordance with Aichi Biodiversity Target 2 of the Strategic Plan for Biodiversity 2011–2020

This indicator calculates the number of countries, which is meaningless at the county scale. No data for this indicator are currently available, and its methodology are still being developed.

- Indicator 15.a.1: Official development assistance and public expenditure on conservation and sustainable use of biodiversity and ecosystems

This indicator shows inter-country assistance and expenditure, especially that provided by developed countries to developing countries. This kind of assistance occurs between the same administrative levels, and county administrative regions generally receive assistance at the provincial or national levels; generally, there is no assistance between county administrative regions. Therefore, the indicator cannot be applied at the county level.

- Indicator 15.b.1: Official development assistance and public expenditure on conservation and sustainable use of biodiversity and ecosystems

The indicator definition is the same as Indicator 15.a.1, and cannot be applied to county-level regions.

- Indicator 15.c.1: Proportion of traded wildlife that was poached or illicitly trafficked

The indicator definition is the same as Indicator 15.7.1, and cannot be applied to county-level regions.

*3.3. Indicator System of Localization*

The indicator system of localization is shown as Table 3:

**Table 3.** Indicator System of Localization.

| Content | Original Indicators | Modality of Localization | Localization of Indicators | | Data Used |
|---|---|---|---|---|---|
| Sustainable forest management | 15.1.1 | A | 15.1.1 Forest area as a proportion of total land area | | Remote sensing images of Landsat |
| | 15.2.1 | R | 15.2.1 Progress towards sustainable forest management | (a) Forest area annual net change rate | Remote sensing images of Landsat |
| | | | | (b) Above-ground biomass stock in forest | – |
| | | | | (c) Proportion of forest area located within protected areas | Remote sensing images of Landsat and vector data of protected areas |
| | 15.4.2 | A | 15.4.2 Mountain Green Cover Index | | Remote sensing images of Landsat |
| Halt and reverse land degradation | 15.3.1 | E | 15.3.1 Status of Land Degradation | (a) Change of land cover types | Remote sensing images of Landsat |
| | | | | (b) Net primary productivity | Remote sensing images of Landsat, meteorological data (temperature and precipitation) and solar radiation data |
| | | | | (c) Carbon storage | Remote sensing images of Landsat and Parameters of carbon pool |
| | | | | (d) Crop yield per unit area | Remote sensing images of Landsat and statistical data on Crop yield |
| Conservation of biodiversity | 15.1.2 | R | 15.1.2 Proportion of important sites for biodiversity that are covered by protected areas | | Remote sensing images of Landsat, meteorological data (temperature and precipitation) and solar radiation data |
| | 15.4.1 | R | 15.4.1 Coverage by protected areas of important sites for mountain biodiversity | | Remote sensing images of Landsat, meteorological data (temperature and precipitation) and solar radiation data |

**Acronyms**: In the Modality of Localization, A is Indicators Adopted; R is Indicators Revised; E is Indicators Extended.

## 4. Results

### 4.1. Indicator 15.1.1

The classified "forest" included three categories: arbor forests, bamboo forests and special shrubs. The accuracy assessment of the land cover types is shown in Table 4.

**Table 4.** Accuracy assessment of land cover types.

|  | 2012 | 2013 | 2014 | 2015 | 2016 | 2017 |
|---|---|---|---|---|---|---|
| Overall accuracy (%) | 78.33 | 88.10 | 85.16 | 86.81 | 82.69 | 77.16 |
| Kappa coefficient | 0.75 | 0.86 | 0.83 | 0.84 | 0.80 | 0.73 |

The Forestry Bureau of Deqing County provided the survey statistics for all categories of woodland area in 2016. These statistical data were used as a reference for evaluating the accuracy of the 2016 forest classification results (Table 5). We used Equation (10) to calculate the accuracy of the forest area as follows:

$$A_A = \left(1 - \frac{|A_1 - A_0|}{A_0}\right) * 100\% \tag{10}$$

where $A_A$ is the accuracy of the forest are in 2016; $A_1$ is the area of classification results; and $A_0$ is the actual forest area from the survey statistics.

**Table 5.** Area accuracy of forest in 2016.

|  | Arbor Forests | Bamboo Forests | Special Shrubs |
|---|---|---|---|
| Actual area (km$^2$) | 137.00 | 209.90 | 58.25 |
| Classification results (km$^2$) | 129.44 | 212.30 | 59.23 |
| Area accuracy | 94.48% | 98.86% | 98.32% |

The distribution of the forests from 2012 to 2017 is shown in Figure 3. The forest area was mainly distributed in western Deqing County. The forest area was 420.75 km$^2$ in 2012. From 2012 to 2017, the forest area showed a slight decrease and then a stable trend. In 2017, the forest area was 407.87 km$^2$.

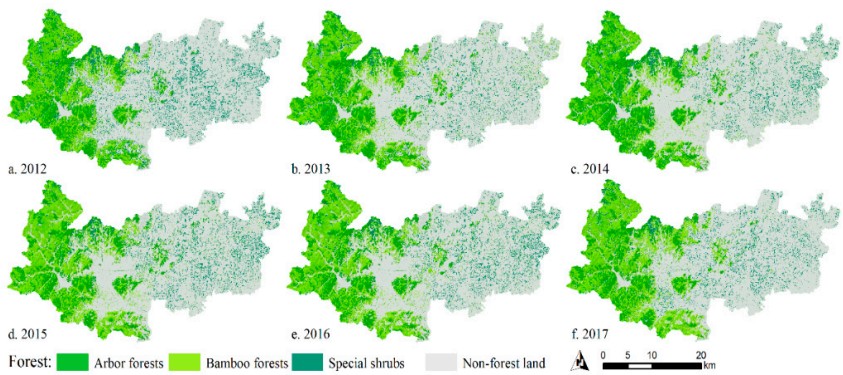

**Figure 3.** Forest in Deqing in 2012–2017.

The protected area covered most of the forest (Figure 4). The proportion of forest area in the protected area was between 76% and 80%.

We used Equation (1) to calculate the proportion of forest area to land area in 2012–2017, which was maintained at approximately 43%. Between 2012 and 2017, the proportion of forest area in protected areas varied between 76.90% and 79.46%, with 78.85% in 2017 (Figure 5).

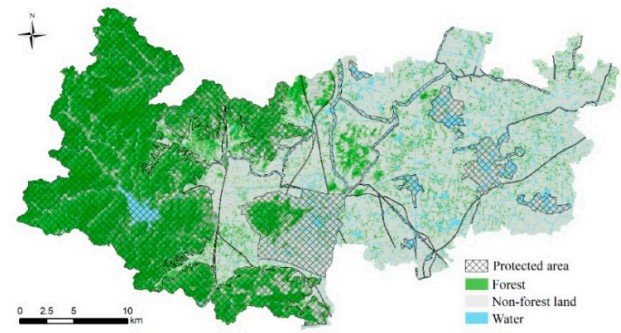

**Figure 4.** Forest distribution covered by protected areas in 2017.

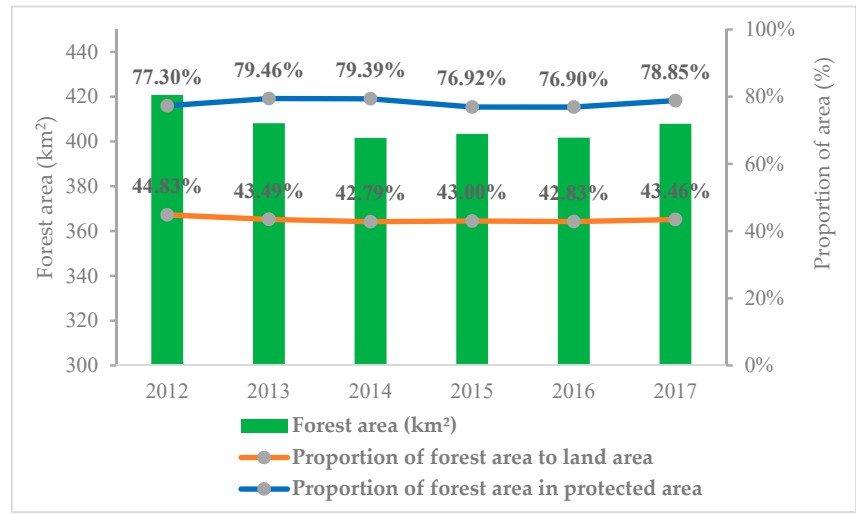

**Figure 5.** Changes in forest area and forest covered by protected areas.

## 4.2. Indicator 15.1.2/15.4.1

The BI (Figure 6) constructed during our research was used to identify important sites for biodiversity. The range of BI was 0–1. Referring to the standard for the assessment of regional biodiversity in China [24] and the distribution of BI in the study area, the biodiversity status was divided into four grades: high (BI > 0.6), moderate (0.4 < BI < 0.6), medium-low (0.2 < BI < 0.4) and low (BI < 0.2). The areas with grades of "high" were the important sites for biodiversity (KBAs).

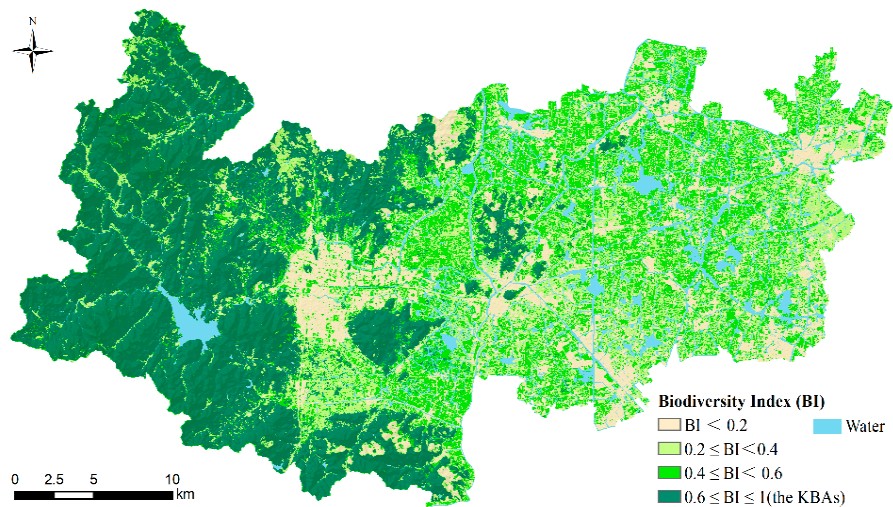

**Figure 6.** Biodiversity Index (BI) in 2017.

Most of the important sites for biodiversity were included in the protected areas. In the mountainous areas, more than 90% of KBAs are covered by the protected areas (Figures 7 and 8).

As shown in Figure 8, between the years 2012–2017, the proportion of key biodiversity areas covered by protected areas varied between 80.68% and 86.96%, with 410 km² in year 2017. The proportion of mountainous areas was stable at approximately 93%.

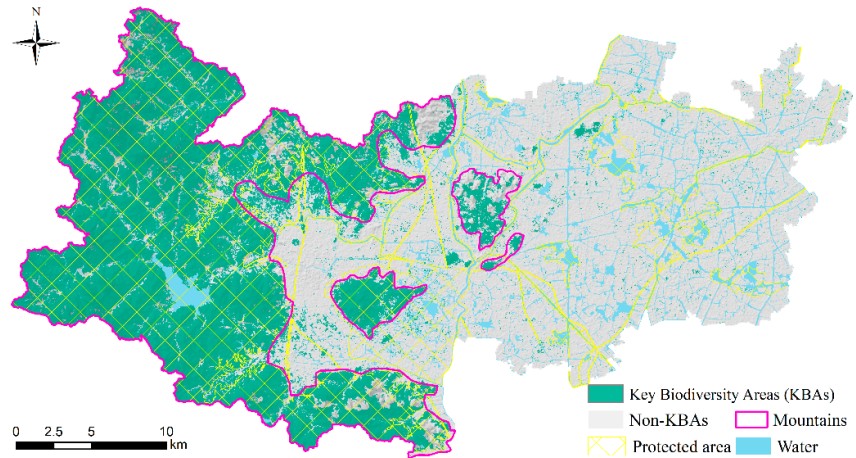

**Figure 7.** Important sites for biodiversity covered by protected areas.

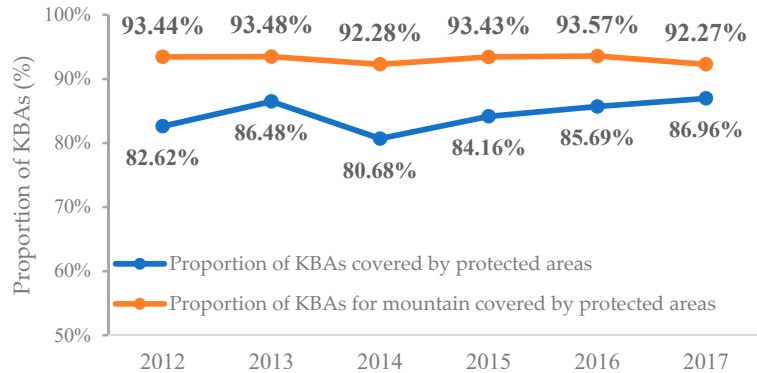

**Figure 8.** Proportion of key biodiversity areas covered by protected areas.

### 4.3. Indicator 15.3.1

- 15.3.1a. Land cover and land cover change

From 2012 to 2017, the land cover change showed a transformation from natural surfaces to artificial surfaces in Deqing County (Table 6). We used the Equation (3) to calculate the proportion of degraded land.

**Table 6.** Change of land cover types.

| Land Ccover Types | 2012 vs. 2017 | | |
|---|---|---|---|
| | Net Change (km²) | Annual Change (km²) | Annual Change Rate (%) |
| NS | −50.03 | −10.01 | −1.22 |
| AS | 50.03 | 10.01 | 8.50 |
| $(P_{N,5})$ | 8.28% | – | 1.53% |

NS, natural surface; AS, artificial surface; $P_{(N,5)}$, the proportion of degraded land in the year of 2012–2017.

Note: Annual change rate (ACR) is calculated as $A = \left( \sqrt[j-i]{\frac{A_j}{A_i}} - 1 \right) \times 100$ [25], where $A_i$ and $A_j$ are areas of each land-cover category at time points I (Time 1) and $j$ (Time 2), respectively.

From 2012 to 2017, the land cover change in Deqing County trended towards land degradation, i.e., the natural surfaces transformed into artificial surfaces, with a net change of 50.03 km$^2$. The land cover change mainly occurs in the central and eastern regions (Figure 9); urbanization was the main reason for the transformation.

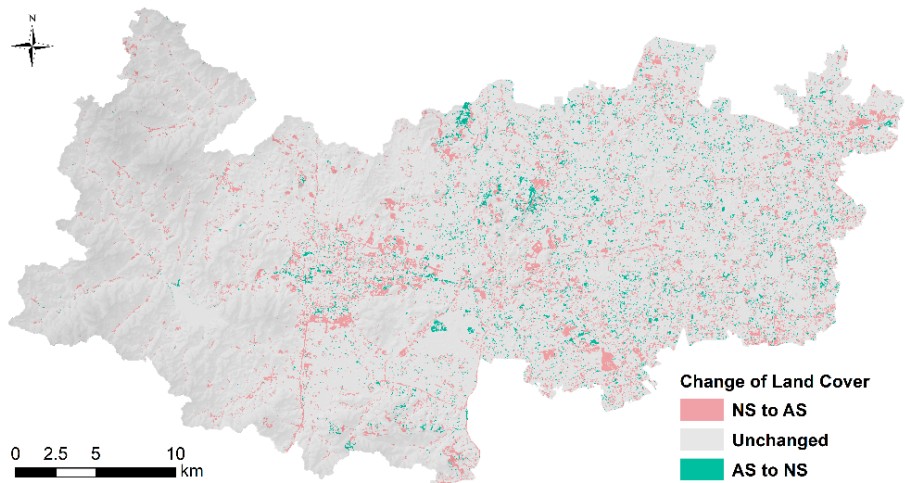

**Figure 9.** Change of land cover from 2012 to 2017. NS, natural surfaces; AS, artificial surfaces.

- 15.3.1b. Net primary production

Using the CASA model to calculate the NPP.

The spatial distribution and variation of NPP are shown in Figures 10 and 11. The net primary productivity of Deqing County was high in the west and low in the east. From 2012 to 2017, the NPP first increased and then decreased. In 2017, the NPP was 420.68 gC/ (m$^2$ × a).

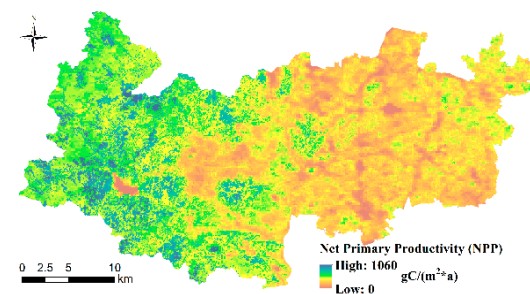

**Figure 10.** Net primary productivity (NPP) in 2017.

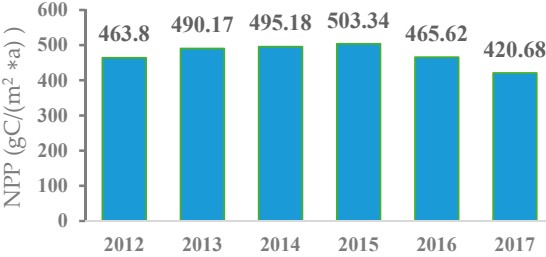

**Figure 11.** Changes in net primary productivity (Regional Average).

- 15.3.1c. Carbon Storage

Using the InVEST model to calculate the carbon storage.

The spatial distributions and variations in carbon storage are shown in Figures 12 and 13. The western region had high carbon stock, while the central and eastern regions had low carbon stock. From 2012 to 2017, the average value of the carbon stock in Deqing County was approximately 80 Mg/hm$^2$, with little change.

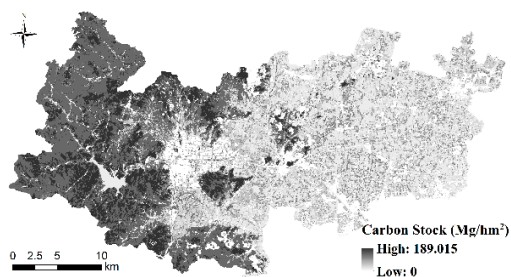

**Figure 12.** Carbon stock in 2017.

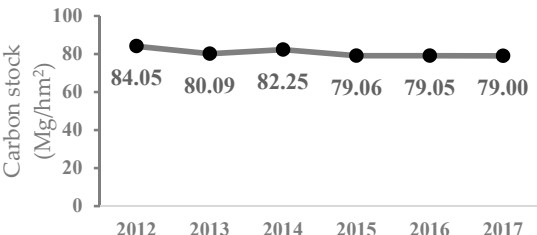

**Figure 13.** Changes in carbon stock (Regional Average).

- 15.3.1d. Crop Yield per Unit Area

The statistical information was combined with geographic information to visualize the crop yields in geographic space.

As shown in Figure 14, cultivated land in Deqing County is mainly distributed in the south and east, and paddy fields are dominant. The crop yield per unit area was maintained at above 7000 kg/hm$^2$ (Figure 15).

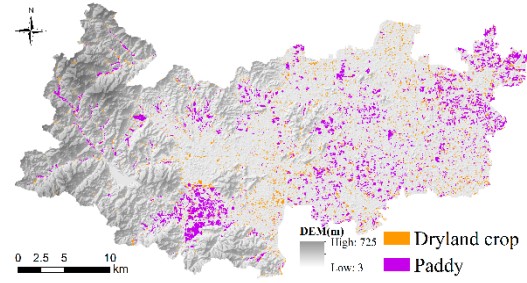

**Figure 14.** Distribution of crops in 2015.

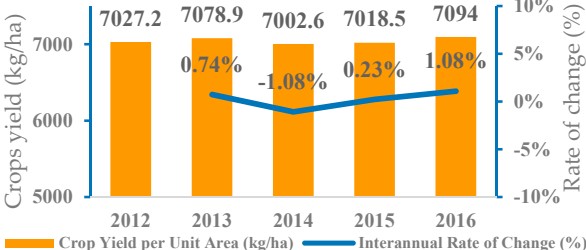

**Figure 15.** Change in crops yield in 2012–2016.

### 4.4. Indicator 15.4.2 Mountain Green Cover Index

We used the Equation (2) to calculate the Mountain Green Cover Index (MGCI).

The spatial distribution of and variation in MGCI are shown in Figures 16 and 17. From 2012 to 2017, the mountain green coverage index of Deqing County was approximately 0.8. In 2017, the MGCI was 0.805.

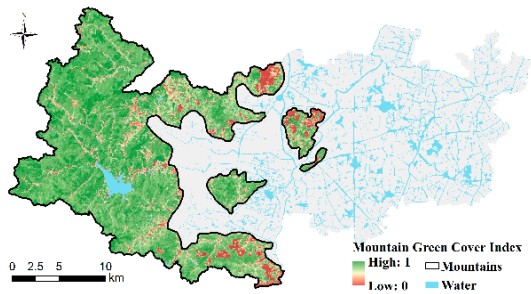

**Figure 16.** Mountain green cover index in 2017.

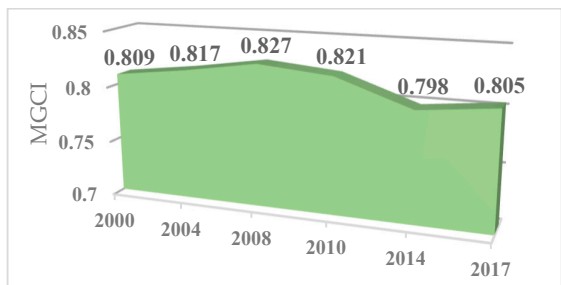

**Figure 17.** Change in mountain green cover index.

### 4.5. Assessment and Practical Progress of SDG 15 in Deqing

According to the traffic light method of the SDGs Index and Dashboard [26], the index values were quantitatively graded. The scores were divided into four segments, i.e., the top quarter (green, basically fulfilling the requirements of the indicator), the second quarter (yellow, to be upgraded), the third quarter (orange, challenging), and the bottom quarter (red, far from achieving the 2030 requirements). "Grey" was used for those indicators that are not comparable. The reference basis for the numerical classifications of the indicators were as follows: I. the SDGs Index and Dashboard; II. China's National Plan on Implementation of the 2030 Agenda for Sustainable Development; III. the world average level; IV. Multi-assessment (without a reference standard above, an appropriate evaluation was conducted with a comparison with national average and position in the world); V: Other (without the above comparison references, the evaluation criteria in the UN SDG metadata were used, or it was "green" if the indicator was in the top quarter of the favourable direction.) (Table 7).

Based on the above evaluation results, Deqing had a good overall performance for SDG 15. Seven of the 10 localization indicators were at the green level after comparison with the criteria, indicating that they have basically fulfilled the requirements of the UN 2030 agenda or seem to be at the forefront of the levels of the country or the world.

(1) Sustainable forest management

Indicators 15.1.1, 15.2.1 and 15.4.2, which were under "sustainable forest management", were rated as "green" and were in a very good state.

**Table 7.** Quantified indicators for SDG 15 in Deqing.

| Content | Localization of Indicators | | Quantitative Result | | Evaluation Reference | | Assessment Level | Trends |
|---|---|---|---|---|---|---|---|---|
| Sustainable forest management | 15.1.1 Forest area as a proportion of total land area | | A | 43.46% | By 2020, China will reach 23.04%. | II | 🟩 | → |
| | 15.2.1 Progress towards sustainable forest management | (a) Forest area annual net change rate | R | 1.54% | Green color band: ≤3% | I | 🟩 | → |
| | | b) Above-ground biomass stock in forest | | – | – | – | ⬜ | – |
| | | (c) Proportion of forest area located within protected areas | | 78.85% | Top quarter | V | 🟩 | → |
| | 15.4.2 Mountain Green Cover Index | | A | 0.805 | Top quarter | V | 🟩 | → |
| Halt and reverse land degradation | 15.3.1 Status of Land Degradation | (a) Change of land cover types | E | 1.53% | | | ⬜ | ↘ |
| | | (b) Net primary productivity | | 420.68 gC/ (m$^2$ ∗a) | | | ⬜ | ↘ |
| | | (c) Carbon storage | | 79 Mg/hm2 | | | ⬜ | → |
| | | (d) Crop yield per unit area | | 7094∗ kg/ha | The average value of China is 5451.9 kg/ha. | IV | 🟩 | → |
| Conservation of biodiversity | 15.1.2 Proportion of important sites for biodiversity that are covered by protected areas | | R | 86.96% | Green color band: ≥ 50% | I | 🟩 | → |
| | 15.4.1 Coverage by protected areas of important sites for mountain biodiversity | | R | 92.27% | The average value of the world is 49% | III | 🟩 | ↗ |

**Note:** Quantitative results are calculated by data in 2017, "∗" is by data in 2016. The trend of quantitative results in the period of monitoring is indicated by arrow direction, upward direction indicates positive development, horizontal direction indicates stability, and downward direction indicates negative development.

The forest coverage of Deqing County was 43.46% (2017), which was much higher than specified in the National Plan of China. In the past six years (2012–2017), the forest coverage had remained high and stable, but the overall forest area and coverage first decreased and then slightly increased, which was directly related to the local construction and development and the transformation of the plantations. Nearly 80% of the forests were in protected areas, which indicated that the ecological functions of most of the forests were very important and had been given attention and protection. The vegetation coverage in the mountainous areas was good, and the mountain green cover index had been maintained at approximately 0.8 since 2000.

In recent years, Deqing has vigorously implemented a series of afforestation projects, such as "precious colour forests" (street trees and landscape forests of precious varieties of trees with different colours), and strived to build a "Green Deqing".

(2) Halt and reverse land degradation

For Indicator 15.3.1, which indicates the objective of "halt and reverse land degradation", the comparable sub-indicator was rated "green".

The crop yield per unit area was much higher than the national average, and the quality of cultivated land was stable and moderate. Carbon storage was relatively stable, but the net primary productivity of the vegetation had a slight downward trend in recent years. The land cover change showed that there was a certain extent of land degradation in Deqing. The natural surfaces (woodland, grassland, cultivated land and water surface) were converted into artificial surfaces (human settlements and unused land) at an annual change rate of 1.53%.

(3) Conservation of biodiversity

Indicators 15.1.2 and 15.4.1 were rated "green" and were in a very promising state.

Important sites for biodiversity (key biodiversity areas, i.e., those that contribute significantly to the persistence of biodiversity) accounted for approximately a quarter of the county area. The KBAs were mainly distributed in western Deqing, showing that the ecological environment in the western areas was of high quality and had rich biodiversity. More than 80% of the KBAs were located in protected areas, while in mountainous areas, the proportion was more than 90%, which is much higher than the world average of 49%. This shows that the vast majority of regions with great significance for biodiversity have received attention and protection.

## 5. Discussion

### 5.1. Clustering and Analysis of Indicators

There were some problems in the SDG 15 indicator framework and indicator design, such as indicator duplication, overly broad definitions and unclear clustering. We analyzed the connotation of SDG 15 and reformed the localization of the indicators.

We considered that the selection and application of the indicators in the SDGs framework should follow the criteria of "relevance", i.e., indicators should be clearly linked with targets in the conceptual framework [27]. However, the existing studies on the application of the SDG indicators neglect or do not pay enough attention to the analysis and clustering of goal connotation [12,28–30]. Hence, the indicators may not conform the theme and purpose of the goal. According to the theme and content of SDG 15, we divided its targets into three clusters (sustainable forest management, halt and reverse land degradation, and conservation of biodiversity). This clarifies the connotation of SDG 15 and is conducive to achieving these targets through the division of labor and the distinguishing of categories. We noticed that there were too many indicators in the original SDG 15 indicator framework, and the relationship between the indicators and targets and the indicators and goal could not be clearly displayed. For example, Indicators 15.1.2 and 15.4.1 indicated the proportion of important sites for biodiversity, which reflects the connotation of "conservation of biodiversity", but the indicators belong to two different targets; Target 15.a reflects the content of the "conservation of biodiversity",

which Target 15.b belongs to the connotation of "sustainable forest management", but the Indicators 15.a.1 and 15.b.1 are exactly the same. In the connotation grouping of SDG 15, Indicator 15.1.1 in Target 15.1, and Indicator 15.4.2 in Targets 15.4, 15.2 and 15.b monitor and indicate the status of and changes in forest vegetation, reflecting the progress of sustainable forest management; Target 15.3 focuses on land degradation and reflects the connotation of "halt and reverse land degradation"; Indicator 15.1.2 in target 15.1, 15.4.1 in 15.4, and Targets 15.5, 15.6, 15.7, 15.8, 15.9, 15.a and 15.c reflect the current situation and labor in "conservation of biodiversity" from different aspects and methods.

Compared with the UN SDGs indicator framework, although the localized SDG 15 indicator system constructed in this paper omitted approximately half of the original indicators, it retained and improved the core and primary indicators. This system also considered that the complex and expensive indicator frameworks would consume excessive manpower and material resources [31], and these indicators could still fully reflect the theme and the three connotations of SDG 15.

### 5.2. Visualizing Spatial Variation Using Geospatial Information

Most SDG indicators result in numbers or indexes, which cannot be used to visualize variability within regions. To show the local details of the spatial distribution of the indicators and to understand the spatial variability of the resource distribution, we visualized the quantitative results of SDG 15 using the geospatial data. In particular, we constructed a remote sensing-based methodological system to identify the KBAs in the Indicators 15.1.2 and 15.4.1. Remote sensing provides an important complementary perspective, revealing details that are often overlooked in official statistics and policy assessments [32]. The remote sensing-based method can continuously and regularly monitor the spatial changes in biodiversity. For Indicator 15.3.1d, we showed the spatial distribution and yield of grain crops by combining statistics with geospatial information. In this paper, a simple visualization method based on land cover type data superimposed with statistical data attributes was adopted; the spatialization of the grain production (spatialization of statistical data) is still a complex problem worth studying.

In addition, the source and acquisition of data in the UN SDGs indicator system has always been the important foundation, but it has been easy to ignore. This paper alleviates this problem to a certain extent. The calculation of SDG indicators requires massive data. In the Metadata document of the SDGs indicator framework of the United Nations, some data sources were recommended for the use of indicators, but these data exist in different channels at all levels, such as through national statistical agencies, data collection organizations or Internet clouds [14]. The data standards and formats are different, and it is difficult to collect and process them uniformly; many of these data cannot be used directly because of statistical calibers and scales. In this paper, geospatial data with high reliability and easy to access, such as remote sensing images and meteorological data, were used to quantify the localization indicators, thus avoiding complicated investigation and the collection of statistical data. The remote sensing data set were uniform in format and easy to obtain and could be a reliable source of SDGs indicator metadata.

### 5.3. Progress towards SDGs in Deqing County

In fact, we must consider the feasibility of measuring the SGIF, taking into account the different geographical and technological levels between regions [22]; the SGIF must be "translated" for specific application areas [28]. The existing research on SDGs indicators focuses on the global, national and urban levels [33–35]; to our knowledge, there are very few studies or reports to quantify the evaluation of grass-roots administrative units (county scale) with a completely localized SDG.

We quantitatively evaluated the progress of Deqing County by using a localized SDG 15 indicator system, which not only clarified the sustainable development of terrestrial ecosystems in Deqing County but also provided an example for similar county-scale assessments in the future. The evaluation results of the case areas showed that the results of "Sustainable forest management" and "Conservation of biodiversity" were promising, with the evaluation grades of "green". In the results of the indices for

"Halt and reverse land degradation", the "crop yield per unit area" was "green", but the "change of land cover types" and "Net primary productivity" showed negative development trends. Human activities and urbanization have changed land cover and land utilization and put pressure on available resources [36]. Rapid urbanization has resulted in greater land utilization and disintegration of natural and man-made features [37], with varying degrees of transformation between natural and man-made surfaces. The development of scenic spots and the construction of roads in villages in Deqing County will lead to increasingly drastic changes in land cover. We recommend that more rational planning be conducted in the process of urbanization in Deqing. In the central and eastern regions, bare land should be fully utilized. In the western mountainous areas, protection should be prioritized, and development should be restricted. At the same time, attention should be paid to the management of the scattered forests in the eastern regions, which can even be transplanted.

*5.4. The Influence of Calculation Method on Results*

Based on the remote sensing image data from Landsat, this paper used the random forest model to classify land cover types and forests in Deqing County. The overall accuracy of the classification was over 80%, and the kappa coefficient was higher than 0.8, except for 2012 and 2017, and the accuracy values in 2012 and 2017 were also close to 0.8. Since the patches of cultivated land and nursery land were relatively fragmented and small, there were "mixed pixels" phenomena, and the overall accuracy was less than 90% due to limitations in image quality and resolution. By overlaying the classification results in 2015 with the vector data of the resource survey (surface coverage) in 2015 provided by the Geomatics Center of Deqing County, it was shown that the two datasets tended to coincide overall and that the spatial distribution was consistent. The results of the forest classification in this paper were approved by the Forestry Bureau of Deqing County. The classification results satisfied the needs of the quantitative evaluation of the indicators. To identify important sites for biodiversity, we constructed a BI based on remote sensing data by referring to the relevant literature [24,38–40]. The distribution of important sites for biodiversity in Deqing County was similar to that of forests. The BI does not directly represent the number of species. From the five sub-indicators of BI, it tended to indicate the diversity of the vegetation and environment and indirectly reflected the distribution of biodiversity through the ecological relationship between each sub-indicator and the connotations of biodiversity. In addition, the InVEST model [17,41] and CASA model [16,42] were used to calculate carbon storage and NPP were widely used and credible models for the existing research. The calculation method for the indicators that was adopted in this paper was not necessarily the best one for each specific parameter; in the present stage of research, we have chosen the commonly used models and methods, focusing on the relevance, accuracy, indication, and timeliness of the small-scale or high-frequency indicator methods [43].

The localization indicator system reformed in this paper mainly uses geospatial data, such as remote sensing images, which currently lack actual data and field verification. Although the geospatial data are easy to obtain and use, if conditions permit, they should still be combined with measured or surveyed data, especially for forest and species inventory data. To make the monitoring results for remote sensing more real and accurate, the measured data should be used as validation and supplements.

**6. Conclusions**

In view of the limitations of the SDGs indicator framework, this paper analyzed and studied the localization reform and application of SDG 15 in the quantitative evaluation at the county level.

(1) In this paper, the connotations of SDG 15 were divided into three groups: sustainable forest management (including Indicators 15.1.1, 15.2.1, 15.4.2 and 15.b.1), halt and reverse land degradation (Indicator 15.3.1), and conservation of biodiversity (including Indicators 15.1.2, 15.4.1, 15.5.1, 15.6.1, 15.7.1, 15.8.1, 15.9.1, 15.a.1 and 15.c.1). We have used four reform methods (adopted, extended, revised and inapplicable indicators) to form a set of localized SDG 15 indicator

systems, which fully utilized the geospatial information and were able to show the internal details and spatial differences of the indicators in county-scale regions.

(2) We applied the localized indicators of SDG 15 to Deqing. Among the quantitative evaluation results of Deqing, many indicators were classified as "green" (in the first quarter of China or the world). The results showed the spatiotemporal distribution and changes of various ecological resources in Deqing and reflected the efforts made by Deqing in the practice of SDG 15, which provided a precise reference for the future planning and development of land ecological natural resources.

The localized indicators can be adapted to case study areas and similar county-level areas. In the process of application, attention should be paid to the acquisition and utilization of geospatial data and the visualization of quantitative results. The research process of this paper provided ideas for analysis and the technical process for the reform and application of other goals in the SDGs. In the transformation of other goals, attention should be paid to the grouped analysis of the connotations of the goals and the applicability of the indicator scale. It should be noted that the SGIF has not yet been fully finalized, and there are still many limitations that need to be improved in practice.

**Author Contributions:** All of the authors designed the study and discussed the basic structure of the manuscript. Conceptualization, Shaoyang Liu, Jianjun Bai and Jun Chen; Formal analysis, Shaoyang Liu and Jianjun Bai; Methodology, Validation and Visualization, Shaoyang Liu; Supervision, Jianjun Bai; Writing—original draft, Shaoyang Liu; Writing-review & editing, Jianjun Bai and Jun Chen.

**Funding:** This research received no external funding.

**Acknowledgments:** We would like to thank the Geomatics Center of Deqing County and Forestry Bureau of Deqing County for providing the data.

**Conflicts of Interest:** The authors declare no conflict of interest.

# Appendix A

**Table A1.** SDG 15 indicator system.

| Goal | Target | Indicator | Tier Classification |
|---|---|---|---|
| 15. Protect, restore and promote sustainable use of terrestrial ecosystems, sustainably manage forests, combat desertification, and halt and reverse land degradation and halt biodiversity loss | 15.1: By 2020, ensure the conservation, restoration and sustainable use of terrestrial and inland freshwater ecosystems and their services, in particular forests, wetlands, mountains and drylands, in line with obligations under international agreements | 15.1.1: Forest area as a proportion of total land area | Tier I |
| | | 15.1.2: Proportion of important sites for terrestrial and freshwater biodiversity that are covered by protected areas, by ecosystem type | Tier I |
| | 15.2: By 2020, promote the implementation of sustainable management of all types of forests, halt deforestation, restore degraded forests and substantially increase afforestation and reforestation globally | 15.2.1: Progress towards sustainable forest management | Tier I |
| | 15.3: By 2030, combat desertification, restore degraded land and soil, including land affected by desertification, drought and floods, and strive to achieve a land degradation-neutral world | 15.3.1: Proportion of land that is degraded over total land area | Tier II |
| | 15.4: By 2030, ensure the conservation of mountain ecosystems, including their biodiversity, in order to enhance their capacity to provide benefits that are essential for sustainable development | 15.4.1: Coverage by protected areas of important sites for mountain biodiversity | Tier I |
| | | 15.4.2: Mountain Green Cover Index | Tier I |
| | 15.5: Take urgent and significant action to reduce the degradation of natural habitats, halt the loss of biodiversity and, by 2020, protect and prevent the extinction of threatened species | 15.5.1: Red List Index | Tier I |
| | 15.6: Promote fair and equitable sharing of the benefits arising from the utilization of genetic resources and promote appropriate access to such resources, as internationally agreed | 15.6.1: Number of countries that have adopted legislative, administrative and policy frameworks to ensure fair and equitable sharing of benefits | Tier I |
| | 15.7: Take urgent action to end poaching and trafficking of protected species of flora and fauna and address both demand and supply of illegal wildlife products | 15.7.1: Proportion of traded wildlife that was poached or illicitly trafficked | Tier II |
| | 15.8: By 2020, introduce measures to prevent the introduction and significantly reduce the impact of invasive alien species on land and water ecosystems and control or eradicate the priority species | 15.8.1: Proportion of countries adopting relevant national legislation and adequately resourcing the prevention or control of invasive alien species | Tier II |
| | 15.9: By 2020, integrate ecosystem and biodiversity values into national and local planning, development processes, poverty reduction strategies and accounts | 15.9.1: Progress towards national targets established in accordance with Aichi Biodiversity Target 2 of the Strategic Plan for Biodiversity 2011-2020 | Tier III |
| | 15.a: Mobilize and significantly increase financial resources from all sources to conserve and sustainably use biodiversity and ecosystems | 15.a.1: Official development assistance and public expenditure on conservation and sustainable use of biodiversity and ecosystems | Tier I/III |
| | 15.b: Mobilize significant resources from all sources and at all levels to finance sustainable forest management and provide adequate incentives to developing countries to advance such management, including for conservation and reforestation | 15.b.1: Official development assistance and public expenditure on conservation and sustainable use of biodiversity and ecosystems | Tier I/III |
| | 15.c: Enhance global support for efforts to combat poaching and trafficking of protected species, including by increasing the capacity of local communities to pursue sustainable livelihood opportunities | 15.c.1: Proportion of traded wildlife that was poached or illicitly trafficked | Tier II |

Note: Tier Classification Criteria/Definitions [44]: Tier 1: Indicator is conceptually clear, it has an internationally established methodology and standards are available, and data are regularly produced by countries for at least 50 per cent of countries and of the population in every region where the indicator is relevant. Tier 2: Indicator is conceptually clear, has an internationally established methodology and standards are available, but data are not regularly produced by countries. Tier 3: No internationally established methodology or standards are yet available for the indicator, but methodology/standards are being (or will be) developed or tested.

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
