# Peer review of "Measuring SDG 15 at the County Scale: Localization and Practice of SDGs Indicators Based on Geospatial Information"

_ijgi, doi:10.3390/ijgi8110515_

Round 1

Reviewer 1 Report

 Measuring SDG15 at County Scale: Localization and Practice of SDGs Indicators Based on Geospatial Information

General comment:

The text of Chapters 1 and 2 needs a thorough rewriting – see below

The entire text requires proofreading.

Chapter 1

It needs to be completely restructured and rephrased:

The SDG 15 and its goals and targets are not properly presented and described

1. The design of the indicator framework does not reflect the goal connotation well row 68

at the moment when the indicators are discussed, they are not used/described – see row 82 – 86, 116 – 119, 139 – 143

The chapter is a mixture of introduction including references and new evaluations County scale is a big scaleif compared to the national scale and not a small one.

Chapter 2

Its name does not reflect the chapter content:

            Analysis of SDG15  x  weaknesses and gaps in the SDG6 and SDG4 indicator

                                                             x Target 12.3

Still the SDG 15 has not been described. Who is author of the Table 1?

Chapter 3- Localization Reform of SDG15 Index

The localization is not properly described in the chapter.

Subchapter 3.2 Localization of Indicators:

                  The localization is not explained

                  The individual indicators are newly attributed but not localized

Row 260- 3.2.3. Indicators revised (R) – Not explained how they are revised

Row 286- vegetation index (EVI) – incorrect name

Row 300– table content is not clear according to relations between column 1 and 2

Rows 260 – 318- The part needs better restructuring and explanations

Row 347, 350, 352- It cannot be applied at the county level - explain why?

Row 354– Table – change the table to a more readable form

Row 357–   rephrase: „high in the west and low in the east“

Chapter 4

Row 355- The Practice of Deqing, China– rephrase the headline

Row 368-9– rephrase the sentences

Row 402– improve quality of the legend text

Row 419– the Figure needs explanation - esp. for the situation in 2015-2016

Row 433- constant 0.5 indicates – rephrase

Row 454– what is the meaning of digital values in the legend of the figure 14

Row 464- The sores???

Row 474– The table needs adaptation – lower fonts, explanation of roman values in the evaluation reference, meanings of colours of the Assessment level

Row 480– how is the green level defined?

Chapter 5

Row 522– It is not clear how you analyzed the existing problems and limitations in the SDGs indicator system

Row 523- Explain how you put forward more effective transformation methods for them.

Row 526– Explain howit retains and improves the core and main indicators

Row 528 - 530– There is no proof of the statement:Indicator system of localization can still monitor and evaluate the level of sustainable development in the environment/biosphere, which is in line with the purpose of UN SDGs design.

Row 541– There is no proof of the following statement since there is no other method to be compared with: This makes the connotation of SDG15 clearer….

Row 547-548Describe how you analyze the types, definitions, contents, calculation methods, data used and evaluation scales of each indicator.

Row 570 – 571Where is a proof of thr statement:but the results of this study are basically consistent with the actual distribution of natural resources in the study area.

Row 579 – 580Where is a proof of the statement:The indicator system of localization and practice of Deqing driven by geographic information data prove that the SDG15 indicator system constructed in this paper is suitable for county scale.

Row 584– rephrase:It makes the evaluation scale of indicators can be refined

Chapter 6

Row 610 – 611The improved indicators can be well adapted to case study areas and similar county-level areas.There is no evidence that the indicators were improved. No comparison were presented.

Row 624- Appendix A title of the Appendix is missing

Reviewer 2 Report

"this paper tries to apply the localization of SDGs to the 144 monitoring and evaluation at county scale by discussing and analyzing the limitations of SDG 15 145 indicator system"

It is the matter of downgrading general data or acquire of more precise data, as well refinement of formulas of indicators. It can be also a matter of harmonization of different acquired or available spatial data and instrumentation of methods. Not only the methodological approach of indicators.

The Authors propose either adopting, extending, revising or ignoring some indicators to make localized judgments and improvements on the indicators, which enhances their applicability. This the matter of belief of readers and the strength of the proof of their research project, that the results reflect the SDG Global Indicator Framework methodology,  which I appreciate.

This also the strong voice in the discussion of methodological aspects of SDGs Global Indicator Framework.

One of the weakness of their discussion is complete lack of the well known problem of MAUP (modifiable areal unit problem) which, as I think, may matter in this context. The readers do not know if the obtained results, values of the particular indicators at the county level should be compared directly to national SDG goals levels or if these results are only part in the set of counties' values (which are either summed up or averaged). 

All above remarks do not depreciate the scientific soundness of the paper.

Reviewer 3 Report

Peer-review for IJGI-589187

The study ”Measuring SDG15 at county scale: localization and practice of SDGs indicators based on geospatial information” by Shaoyang Liu et al. utilizes various geospatial data in an attempt to quantify progress towards meeting sustainable development goals (SDG) set by the United Nations. The main question addressed in the study is to test the measurability of the progress at a county scale (Deqing county, China), and hence to test whether the indicators developed to measure the progress are applicable at this administrative level. The main findings of the study state that with a slight modification, the indicators can be used to track progress towards the goals of SDG15 at the studied level, and that geographic information is a powerful tool in expressing spatial variability in the progress. The problem posed in the manuscript is relevant, and the study is a welcomed attempt to overcome the problem.

That said, I have several concerns that should be considered, and some suggestions for improvement. First and foremost, I’m not convinced that the results actually support the main conclusion that the modified indicator system is suitable for measuring the progress towards the goals of SDG15. Six out of the ten indicators were quantified using remotely sensed records. However, the manuscript offers no details of the accuracy of the quantification except for laconically stating that post-processing was carried out. Secondly, due to the complex structure of the manuscript, I find it very difficult to follow. I listed the most important shortcomings with more detail, as well as some suggestions as how to overcome them, below.

Major issues

1) Introduction: I would like to see a clearer justification for the study, and the logical flow leading to the research questions needs to be improved. For example, the progress towards the SDGs set by the UN have been monitored and measured at within-national levels elsewhere. Hence, the introduction should consider how successful have these attempts been, what have been major the stumbling blocks, and why might the situation differ in Deqing county? Furthermore, the amount of repetition in the introduction section must be reduced. For example, the lines 54–67 mostly repeat what was said before, as do lines 126–134. Please, remove the excessive repetition and replace it with a logically proceeding justification of the study.

2) Methods: There is recent and older evidence that due to, e.g., changes in data quality, indices quantified from remote sensing material are prone to bias and random error that might mask or intensify the actual changes. Similarly, for similar reasons, the accuracy of land cover classification may vary. How these errors influence the obtained results needs to be addressed in the manuscript. For example, state how successful your land cover classification and forest cover estimations were (e.g., errors of omission and commission, kappa coefficient etc.), and how well the InVEST-model estimated aboveground carbon storages. I would also like to see some more explicit estimates of species diversity (other than estimating area of potential habitat) used in the analysis. Could these be added to the manuscript?

3) Methods: The quantification procedure of some of the used indices, such as habitat quality index, were not provided in sufficient detail to assess their suitability to meet the aims of the study. Furthermore, a rationale for how the indicators “change of land cover types” and “crop yield per unit area” were quantified was completely missing from the manuscript. Please, provide the necessary information of how all the used indices were quantified, what data were used in the quantification, and how the data needed to be processed to achieve the quantification. This information can partially be put to appendix if it does not fit to the manuscript itself.

4) Discussion: The discussion section does not place the study in context with another similar studies. Instead, the section mostly lists what was done, and reviews the results. Currently, the discussion section contains only one reference to other studies! This needs to be improved. Consider how your results align with the other studies that have explored the progress towards SDGs. Similarly, I would welcome discussion on the accuracy of the land cover classification, forest cover estimation and InVEST model results suggested in point 2 above.

5) Whole manuscript: I urge the authors to increase the clarity of the manuscript by first organizing the paper to follow the classical IMRD-system more closely. For example, the results-section starting from line 356 includes a lot of material that belongs to the methods-section (such as the equation at lines 425–428. Please, ensure that all manuscript parts are in their proper places, and that the narrative of the manuscript progresses in a logical manner. Secondly, the clarity of the methods-section must be improved. Please, present clearly what material was used, how the material was processed, and what was the question that the particular combination of material and its processing aimed to answer. Similarly, the study are description should be placed in one place, not scattered to introduction, materials and methods, and results.

Smaller issues:

All figures and tables: All figures and tables lack references in the text. Place, insert references to the figures and tables to appropriate places. Fix the figure numbering to start from 1.

Figure 6: Some attributes show in the map may be visualized in two different classes. For example, is an area that has a biodiversity index of 0.2 colored as light yellow or green? Clarify the figure legend.

Abstract: I recommend to define the acronym SDG already in abstract (or even in the title) as it might be unclear to some readers.

29: Replace “environmental beauty” with “environmental sustainability”. 42–46: Complicated sentence, clarify. 49: Replace “More realistically” with “In practice”. 50: Region can refer to an area with practically any spatial scale. Thus, the term is ambiguous. Replace “region” with, e.g., province or other administrative level. 54–67: Repetition, remove. 75: Clarify what is meant by “core key indicators”? 93: Remove the citation just before [1] 104–107: Repetition, remove. 126–128: The sentence is difficult to understand. For example, what is meant by “moving transcending the cockpit-ism?” Make the statement more clear or remove. 130–134: Repetition, remove. 137–139: Repetition, remove. 144–145: You mostly use acronym SDG15, but here you use SDG 15. Choose either and be consistent throughout the manuscript. 151–160: I recommend introducing the study area in materials and methods. 157–160: You can mention that the area is undergoing rapid industrial development, but I advise to refrain from using phrases such as “have sprung up like mushrooms after a spring rain”. 164–165: Remove author names before citations [18] and [19]. 183–184: Repetition, remove. 186: Replace Target with Targets. 193: The targets can be divided by whom? The authors, The UN, other scholars? Please, clarify. 230: Specify what is meant by “widely used and relatively mature models and methods”. 235: Clarify what is referred with “it can be divided..”? 268: Spell out the acronym KBA at the first time of appearance. 271–272: Construction land, or the presence of artificial land covers in general do not automatically decrease biodiversity. For example, railway verges may locally increase the diversity of vascular plants. Modify. 275: Clarify what is meant by “each endangered level”. 289–291: Similar to genetic diversity, species diversity cannot be directly monitored by remote sensing. Instead, similar to genetic diversity, you are quantifying surrogate measures for species diversity. Rephrase. 296: Number this and the sequential equations. 324–330: Why should administrative borders at national scale be any better for understanding species diversity than those at smaller spatial scales (such as province boundaries)? 354, Table 5: I suggest adding the information source for each index to this table. E.g., whether the index was quantified using Landsat data, meteorological data, or something else. 357–366: I suggest moving the study are description to material and methods-section. 368–375: This needs to be considerably improved. What were the data sources, what were they used for etc? See comment 3 above. 378–383: The random forest model and its outcome, as well as the PCA need to be explained with more detail. See comment 3 above. 438–441: Provide more details on the parametrization of the InVEST model, or cite sources that give a more detailed explanation. 481–482: I’m not convinced of this statement. Replace “are obviously” with softer statement, e.g. “seem to be”. 496–497: This conclusion cannot be drawn based on your results. Rephrase or remove. 503–508: Move this to discussion. 514: Your results indicated that the western parts of the study had higher forest cover compared to the eastern parts. While this suggests that these parts also have higher biodiversity compared to the east, this cannot be verified based on your results. Rephrase the statement. 517–519: Irrelevant, remove. 528: I would say that the framework has changed considerably. Rephrase the statement. 558: Replace “unfairness” with “spatial variability”. 570–571: What is this statement based on? Clarify or remove. 577: What did these “few studies” find out? How where they comparable to your results? 585–586: I suggest to soften the statement “complete study and quantification of..”. 600–602: Instead of concluding, this statement repeats what was done in the study. Rephrase. 603–604: I do not agree with the point that applying an indicator system validates the system. Such validation would require testing for the accuracy of the indicator quantification. Rephrase.

Reviewer 4 Report

The work done by authors is an interesting and new concept. However, some suggestions are proposed for further improvement.

Major comments

Methodology and results:

LN 369 “13 multi-spectral remote sensing images of Landsat 7 ETM+, Landsat 8 OLI sensor”. The data selection method is not scientifically sound, and clear explanations regarding data sources and method are required. I have some questions as follows,

Why April to November? Why only 13 images? What are the sources Landsat 7 ETM+, Landsat 8 OLI? What would be the could cover?

Please refer the following sources for more information regarding data and data sources  https://www.mdpi.com/2225-1154/7/8/99/htm

In Figure 3, there is only one processed image in each year from 2013-2017 (six outputs), but you have used 13 Landsat images. If you used more than one image in each year, what is the method that you have incorporated to combine images in R or other software? Explanation of the process is primarily required. Finally, it is suggested to add R code to the manuscript as an annex.

LN 383, If you have identified three types of forest (arbor forests, bamboo forests and special shrubs), those categories should add to Figure 3 or any other maps. Figure 3 shows only forest, non-forest, and water. Other than this, what is the method that you have used to classify the water and non-forest because you mentioned that classification was carried out for the identification of three types of the forest? As an over role aspect, there are serious problems in the methodology. Hence, strong method with flowchart is recommended.

Classification accuracy: Classification accuracy assessment should is required. Otherwise, results of the classification will not trust. You can use R software for both classification and accuracy assessment.

LN 398, What is the method that you have used for grading the BI from high to low?

LN 406, “4.2.3. Indicator 15.3.1a Change of land cover types”. The employed method for change of land cover types in 2012-201 is not meaningful, and way of results presenting is not appropriated (Figure 9). You can use change detection matrix to show the temporal changes    

What is the method or theory that you have accomplished for the declaration of forest area as key biodiversity area?

What is the source of Carbon data?

What is the source of crop yield data?

LN 455, In this process, first you have to calculate the NDVI. What is the Landsat data that you used for the calculation because you have mentioned 13 Landsat data sources?

LN 453-455, I supposed that you have used only paddy yield. If so, why use as a grain? Other than this, in Figure 14 shows only 2015 but Figure 15 shows 2012-2016. If you have data only for one year, use it for both.    

Discussion:

As an over role aspect, the discussion should mainly emphasize the argument or conversion of the results that you have gained. However, the beginning point of the discussion section and section 5.1 slightly explain the process and method but other two sections are quite appropriate. Finally, it is suggested to improve the discussion in more scientifically.

Conclusions:  

It seems to be conclusions is too long. You can improve the last paragraph by summarizing the key points of the first four paragraphs.    

Minor comments

Equations:

LN 388, Equation caption is required. LN 415, Equation caption is required. LN 425-428, Equation caption is required. LN 445, Equation caption is required.

Figures:

Figure 1 is not included; Figure 2 should be changed as Figure 1 General color ramp for the elevation is white to brown not white to green. Suggested to change Figure 2 DEM color. Three-section of Figure 2 should be named as a, b, and c. and updating of image caption is required. Figure 5: Image resolution is not adequate and labeling of x and y axis are required. Same issues are found in Figure 8, 9, 11, 13, 15, and 17 All maps are not adequate resolution and legend is not clear

Summary

The manuscript is needed to thoroughly improve by referring the scientific research papers. The main issue is noticed in the methodology section including data sources and work flow.     

Round 2

Reviewer 1 Report

Dear authors,

I appreciate very much the significant correction you have done.

I agree with your solutions and changes, which were done according to my first review.

The attached file is a list of details and notes to correct mistakes, typos, etc

Author Response

Thank you.

Reviewer 3 Report

Comments in the attached pdf-file.

Author Response

Thank you.

Reviewer 4 Report

The manuscript has been improved based on the given comments, but some minor comments are suggested.

Less could is not scientific. What is its limit? You can mention it as less than some amount. “Which indicates that the biodiversity in forest area may be high” this is too general. Can you add some reference to prove your statement? General color ramp for the elevation is white to brown not white to green. The same issue has been detected in the previous time also. labeling of x and y-axis are required for all graphs. The same issue has been detected in the previous time also. Why don’t you add workflow diagram? The same issue has been detected in the previous time also. Format of the reference 25 is incorrect and it should be correct as follows.

MDPI and ACS Style

Dissanayake, D.; Morimoto, T.; Ranagalage, M.; Murayama, Y. Land-Use/Land-Cover Changes and Their Impact on Surface Urban Heat Islands: Case Study of Kandy City, Sri Lanka. Climate 2019, 7, 99.

Author Response

Thank you.

Round 3

Reviewer 3 Report

My comments can be found from the attached file.

Author Response

Thank you.
